# New Insights into the Hygroscopic Character of Ionic Liquids: Study of Fourteen Representatives of Five Cation and Four Anion Families

**DOI:** 10.3390/ijms25084229

**Published:** 2024-04-11

**Authors:** Esther Rilo, Alejandro Rosende-Pereiro, Montserrat Domínguez-Pérez, Oscar Cabeza, Luisa Segade

**Affiliations:** 1Departamento de Física, Facultade de Ciencias, Universidade da Coruña, Campus da Zapateira, 15071 A Coruña, Spain; esther.rilo.siso@udc.es (E.R.); alejandro.rosende@academicos.udg.mx (A.R.-P.); montserrat.dominguez.perez@udc.es (M.D.-P.); oscar.cabeza@udc.es (O.C.); 2Departamento de Estudios para el Desarrollo Sustentable de Zonas Costeras, Universidad de Guadalajara, Gómez Farias #82, San Patricio-Melaque 48980, Jalisco, Mexico

**Keywords:** ionic liquid, hygroscopicity, atmospheric moisture, absorption, adsorption, desorption

## Abstract

Over the past three decades, the synthesis of new ionic liquids (ILs) and the expansion of their use in newer applications have grown exponentially. From the beginning of this vertiginous period, it was known that many of them were hygroscopic, which in some cases limited their use or altered the value of their measured physical properties with all the problems that this entails. In an earlier article, we addressed the hygroscopic grade achieved by the ILs 1-ethyl-3-methylimidazolium chloride, 1-ethyl-3-methylimidazolium bromide, 1-ethyl-3-methylimidazolium methyl sulfate, 1-ethyl-3-methylimidazolium ethyl sulfate, 1-ethyl-3-methylpyridinium ethyl sulfate, 1-ethyl-3-methylimidazolium tosylate, 1-ethyl-3-methylimidazolium tetrafluoroborate, 1-butyl-3-methylimidazolium tetrafluoroborate, 1-dodecyl-3-methylimidazolium tetrafluoroborate, 1-butyl-3-methylpyridinium tetrafluoroborate, 1-butyl-1-methylpiperidinium bis(trifluoromethyl sulfonyl)imide, 1-methyl-1-propylpyrrolidinium bis(trifluoromethyl sulfonyl)imide, 1-butyl-1-methylpyrrolidinium bis(trifluoromethyl sulfonyl)imide, and methyl trioctyl ammonium bis(trifluoromethyl sulfonyl)imide. The objective was to determine the influence of the chemical nature of the compounds, exposed surface area, sample volume, agitation, and temperature. For this purpose, we exposed the samples to abrupt increases in relative humidity from 15 to 100% for days in an atmosphere chamber and then proceeded with the reverse process in a gentle manner. The results show that the sorption of water from the atmosphere depends on the nature of the IL, especially the anion, with the chloride anion being of particular importance (chloride ≫ alkyl sulfates~bromide > tosylate ≫ tetrafluoroborate). It has also been proven for the EMIM-ES and EMIM-BF_4_ samples that the mechanism of moisture capture is both absorption and adsorption, and that the smaller the exposed surface area, the higher the ratio of the mass of water per unit area.

## 1. Introduction

Ionic liquids (ILs) are materials first described at the beginning of the last century, but their real boom came in the 1990s when the synthesis of new ones and their applicability began to grow exponentially. Numerous papers have been published on the synthesis, characterization, and applications of new ILs. This immense volume of scientific and technological data has led to greater knowledge and dissemination of the extraordinary physicochemical properties of these materials, and to their implementation in a multitude of applications of very diverse natures and with very promising results [1,2]. One of the physicochemical characteristics of most of these materials, and a priori not the most desirable, is their hygroscopicity, which strongly affects their physical properties [3,4,5,6,7,8,9,10,11,12]. This occurs not only in hydrophilic ILs, but also in hydrophobic ones, which take water from atmospheric moisture [13,14,15,16,17]. Considering that the magnitude of many physical properties strongly depends on the water content, even in very low quantities [9,12], the contact of ILs with the atmosphere can result in the misfunction of a device or a process based on these materials. In addition, this characteristic can be useful for some applications, such as the use of ILs as hygrometers [18] or desiccants [19] for the removal of moisture from air or other gases [20,21,22] due to their high hygroscopic character. Compared to conventional desiccants, which are characterized by being corrosive and energy consuming, ILs can provide a better performance in this sense, can present better recyclability, and are less susceptible to crystallizing at high concentrations [19].

Among the goals that our research group has been pursuing in recent years has been to investigate the effect that the presence of water in ILs produce in their physical state or in their physical properties. Regarding the former, we have published how the absorption of ambient humidity causes samples of 1-ethyl-3-methylimidazolium octyl sulfate and 1-ethyl-3-methylimidazolium decyl sulfate to be reversibly transformed into rigid hydrogels [23,24]. It was also demonstrated that the properties of these hydrogels underwent a drastic change, showing practically zero fluidity but relatively high electrical conductivity, which is contrary to the accepted charge transport model in ionic liquids. Further concerning the effect on the physical properties, we carried out an exhaustive study of the influence of the presence of small quantities of water in ethylammonium nitrate and propylammonium nitrate on density, viscosity, conductivity, surface tension, and the refractive index or even on electrochemical windows [12]. It was demonstrated that the transport properties were particularly affected, such that the average percentage change was between 39 and 56% for water contents up to 30,000 ppm.

These recent objectives were the direct consequence of a previous work [7] in which we largely tested the degree of hygroscopicity of ILs based on the imidazolium cation and the alkyl sulfate and tetrafluoroborate anions. We evaluated how the length of the alkyl chain in the anion and cation, specifically in the sulfate anion or imidazolium cation, affected the hygroscopicity of these materials. To the best of our knowledge, this was the first paper published with this type of measurement applied to ILs. In fact, very few papers have been published referring to the quantification of the hygroscopic character of ILs [4,7,11,14,16,17,25,26,27,28,29,30,31,32], and a few others have also explored and optimized methodologies to detect water uptake from air in ILs, such as near infrared [25], RMN spectroscopy [9], volumetric analysis and the vibrational spectroscopy of acoustically levitated droplets [27], cathodic stripping voltammetry [26], gas chromatography [33], and gravimetry [4,7,13], this last being the most widely used. A classification with four levels of ILs regarding hydrophilicity was suggested: super-high, high, medium, and low, according to their water sorption capacity [14]. In addition, the hygroscopic character of ILs has been an object of study in theoretical [34] and computational papers [15,35].

There are still many questions to be elucidated about the hygroscopic characteristics of many other non-investigated ILs, such as whether the water is absorbed or adsorbed, the reversibility of the process, the quantity of water that a given IL captures, the time of saturation, and the influence of the surface area of the IL exposed to the atmosphere, taking into account that not all ILs exhibit the same behavior. Some questions have been partially clarified in the literature, such as the role of the anion and cation, the influence of the chain length, and some insights into the capacity of absorbing water from moisture. The most important question to be clarified is the mechanism of water sorption, and in this sense, three different models were proposed in ref. [14] but did not agree with computer simulation findings [15]. One of the aims of this study was trying to clarify this question for some of the most hygroscopic ILs.

With these antecedents, in this work, we continued our line of research on hygroscopicity by focusing on the study of new families of ILs and how their chemical nature influences the capture of environmental moisture, analyzing the role played by cations and anions. To this end, experiments carried out in an atmosphere chamber are shown in which air was saturated from 15 to 100%, in one or two steps, and the reverse process was followed (from 100 to 15%). In addition, studies on the influence of the surface area exposed to air, sample volume, agitation, and room temperature are presented. The ILs investigated in this research were 1-ethyl-3-methylimidazolium chloride (EMIM-Cl), 1-ethyl-3-methylimidazolium bromide (EMIM-Br), 1-ethyl-3-methylimidazolium methyl sulfate (EMIM-MS), 1-ethyl-3-methylimidazolium ethyl sulfate (EMIM-ES), 1-ethyl-3-methylpyridinium ethyl sulfate (EMPY-ES), 1-ethyl-3-methylimidazolium tosylate (EMIM-Ts), 1-ethyl-3-methylimidazolium tetrafluoroborate (EMIM-BF_4_), 1-butyl-3-methylimidazolium tetrafluoroborate (BMIM-BF_4_), 1-dodecyl-3-methylimidazolium tetrafluoroborate (dDMIM-BF_4_), 1-butyl-3-methylpyridinium tetrafluoroborate (BMPY-BF_4_), 1-butyl-1-methylpiperidinium bis(trifluoromethyl sulfonyl)imide (BMPIPE-TFSI), 1-methyl-1-propylpyrrolidinium bis(trifluoromethyl sulfonyl)imide (MPPYRR-TFSI), 1-butyl-1-methylpyrrolidinium bis(trifluoromethyl sulfonyl)imide (BMPYRR-TFSI), and methyl trioctyl ammonium bis(trifluoromethyl sulfonyl)imide (MtOAM-TFSI). We chose ILs based on the same cation, EMIM, which is one of the most studied, but we also studied the behavior of others based on imidazolium cations with different alkyl chains, as well as pyridinium, pyrrolidinium, piperidinium, and ammonium cations, all of which are representative of the most widely used IL families.

## 2. Results

As introduced in the previous section, several experiments were conducted to study the hygroscopicity of different families of ILs according to different criteria. Thus, in the following subsections, the influence of the chemical nature, surface area exposed to air, volume of the samples, agitation of the samples, and small changes in the ambient temperature on the capture of ambient moisture by the ILs were studied. In all the experiments carried out, there was a target sample (an empty plate without any sample) to verify that water absorption was due to the presence of ILs, and not condensation on the glass plate.

### 2.1. Influence of Chemical Nature

The first experiment was performed by measuring the mass variation (Δm) with time (t) experienced by ten samples of ILs abruptly exposed to 100% relative humidity from a nearly dry atmosphere with only about 15% relative humidity. The ILs studied were EMIM-Cl, EMIM-Br, EMIM-MS, EMIM-ES, EMIM-Ts, dDMIM-BF_4_, MPPYRR-TFSI, BMPYRR-TFSI, BMPIPE-TFSI, and MtOA-TFSI, and were kept in an atmosphere chamber. The procedure followed is described in detail in Section 4.2. as number 1. Specifically, Figure 1a shows the mass variation experienced by the ten samples with approximately 1 g of mass contained in Petri plates of 3.9 ± 0.2 cm in diameter at room temperature and atmospheric pressure. The quantity of the IL used was sufficient to cover all the Petri plate surfaces. Obviously, the measured Δm corresponded to the water adsorbed or absorbed from the atmospheric moisture. The reverse process, i.e., the decrease in mass over time, which the ten ILs experienced as the relative humidity of the chamber air decreased abruptly from 100% to the initial humidity of 15%, is shown in Figure 1b. In this case, the procedure followed was procedure number 2, described in Section 4.2.

The second experiment involved ILs based on imidazolium and pyridinium cations with two common anions to elucidate the influence of the cations and anions on the water absorption process. Samples of approximately 1 g of BMIM-BF_4_, EMIM-ES, EMPY-ES, and BMPY-BF_4_ were poured into 3.9 ± 0.2 cm diameter Petri plates following the procedure described in Section 4.2. with number 3. Therefore, the relative humidity of the chamber where the samples were deposited was increased in two steps, from 15% to 68% and from 68% to 100%, always allowing saturation for 24 h at each humidity grade. The results of the increments in mass versus time at room temperature and atmospheric pressure are shown in Figure 2a. In this experiment, the humidity decrease process was again performed in a single stage, from 100% to 15%, and the results of Δm vs. t are shown in Figure 2b.

### 2.2. Influence of Surface Area Exposed to Air

The third experiment was designed to elucidate the dependence of the hygroscopic grade of the ILs on air-exposed surfaces. For this purpose, Petri plates of six different diameters containing the same IL, EMIM-ES, were used in amounts of 0.08 mL/cm^2^. The surfaces exposed to air in ascending order were 13.14, 20.11, 37.83, 61.38, 95.51, and 149.14 cm^2^, and the initial temperature and relative humidity were 18 °C and 15%. In this case, the procedure, described in Section 4.2. as number 1, consisted of abruptly increasing the relative humidity from 15% to 100%. The results obtained for the time-dependent mass variation (due to the water adsorbed from atmospheric moisture) are shown in Figure 3a, while Figure 3b presents the mass of water captured per square centimeter (Δm/S, where S is the surface area). As observed, the curves did not collapse to a common one; however, if we also consider the plate perimeter, all curves merged into a common one, as described below.

### 2.3. Influence of Volume

The evaluation of the influence of IL volume was carried out using two series composed of three EMIM-ES and three EMIM-BF_4_ samples of 1.0, 1.5, and 2.5 mL, all on a Petri plate 3.9 ± 0.2 cm in diameter. The third experiment followed procedure number 1, described in Section 4.2., in which the relative humidity was increased from 15% to 100% and the temperature was maintained constant at 22 °C. The results are presented in Figure 4, specifically the variation in mass, that is, the adsorbed water content, with time for the six samples.

### 2.4. Influence of Agitation and Temperature

As shown in Figure 4, the water adsorbed depended on the depth of the IL in the Petri plate, and not only on the surface area exposed to humidity for EMIM-ES, but much less for ILs with the BF_4_ anion. To explain this fact, we must conclude that some of the absorbed water entered the bulk of the liquid, while the other water was deposited on its surface forming a film, being a different proportion for the two anions studied. If this is so, for a stirred mixture, where water is uniformly distributed throughout the bulk of the sample, the drying process must be different for both anions. To verify this, different aqueous solutions of approximately 50% in weight with the EMIM-BF_4_, BMIM-BF_4_, and EMIM-ES ILs were prepared and gently stirred. About 1 mL of each mixture was placed on 3.9 ± 0.2 cm diameter Petri plates, which were stored in a dry chamber with approximately 15% humidity at room temperature. While the EMIM-BF_4_ and BMIM-BF_4_ mixtures did not appreciably change the original mass after 24 h (i.e., they did not lose the water mixed in the bulk), the EMIM-ES with about 50% in mass of water did, and 24 h later, the water content of the mixture was approximately 2% in mass, as verified by Karl Fischer measurements. Therefore, EMIM-ES lost approximately 96% of the water mass it originally had, even for the stirred mixture, indicating that water molecules can move from the bulk to the surface.

Another experiment was performed to observe the influence of ambient temperature on the water absorption process. To do this, we left the same sample of EMIM-ES twice at two different constant temperatures in the chamber saturated with moisture. After 120 h, the mass of absorbed water ranged from 1.409 g at 17.5 °C to 1.812 g at 22.0 °C, with a temperature of approximately 0.5 °C.

## 3. Discussion

### 3.1. Influence of Chemical Nature

As previously indicated, most ILs are substances that have a strong tendency to capture water from the atmosphere, and this tendency fundamentally depends on the nature of the anion that composes them, as the influence of the cation is much smaller, as described in previous works [7,14,28]. The extent to which they absorb or adsorb requires an in-depth study of the behaviors of different families of ILs, such as those presented in Figure 1a, where it is observed that upon saturating the environment with humidity, the ILs with the TFSI anion did not show appreciable mass variation, independent of the three cations used: pyrrolidinium, piperidinium, and ammonium. After 21.3 h, we analyzed all the ILs that increased in mass and observed that dDMIM-BF_4_ was the one that did so in a more moderate way (approximately 26%), followed by EMIM-Ts and EMIM-Br, which experienced increases of 62% and 67%, respectively, with respect to their initial masses. These were closely followed by the two sulfates with a 71% increase, ending the series with EMIM-Cl, which had the highest percentage of water absorbed from the atmosphere at 94%. Between 24 and 72 h from the beginning of the experiment, the ILs based on halide and sulfate anions maintained similar behaviors, but from the third day on, differences between them began to be clearly distinguished. At 142 h, at the end of the experiment for the halides, there was an increase in mass of 2.196 g and 1.946 g for chloride and bromide, respectively, which represented 215% and 141% of the water absorbed with respect to their initial masses. For the sulfates, the experiment was extended up to 236 h, reaching mass increments of 2.422 g and 2.154 g for methyl sulfate and ethyl sulfate, respectively, representing increases of 192% and 171%, respectively. Thus, we can conclude that the hygroscopic grade depends on the nature of the IL and that water capture is a slow and long process, taking more than 10 days to saturate the sample in a 100% humidity atmosphere at room temperature. A significant part of the captured water migrates into the inner part of the sample, but other molecules form a transparent water film on the IL as previously proposed, both experimentally [7,14,28] and theoretically [34].

As shown in Figure 1b, the water desorption process was much faster than the moisture capture one, since at 40 h in a dry chamber (with a humidity grade of approximately 15% at room temperature), the six hygroscopic ILs practically lost the mass of the water captured, stabilizing their weights at those corresponding to the pure ILs. It should be noted that none of the hygroscopic ILs reached the initial mass value, conserving from 7% to about 18% of the excess mass in respect to the pure state, which suggests that the moisture uptake process could not be reversible absorption, with some part of the water molecules remaining bound in the IL network inside of the bulk.

Two of the most widely used families of ILs are those based on imidazolium and pyridinium cations, which is why we focused on their hygroscopic behavior through EMIM-ES, EMPY-ES, BMIM-BF_4_, and BMPY-BF_4_. Note that the water captured by both families of ILs was very different, probably due to the hydrophobic character of BF_4_ and the hydrophilic nature of ES. In Figure 2a, it can be clearly seen that the anion contributed the most to the water uptake process, since sulfate multiplied by five times the amount of water captured with respect to tetrafluoroborate at a relative humidity of 68%. In contrast, for the same anion, the imidazolium cation presented the highest value of water absorption, ranging from 10 to 27%. On the other hand, Figure 5 shows another view of the saturation process in the atmosphere chamber containing these imidazolium and pyridinium samples. Herein, we present the water absorbed at different humidity grades after the sample was left for 24 h. It shows that at up to 55% humidity, the BF_4_-based ILs absorbed only a low water quantity, which reached only 11–13% of the total mass increase at saturation, whereas for the sulfates, only 5% of the mass was absorbed at saturation (which represented a mass increase of 3% with respect to the initial mass of the ILs with BF_4_ and 6% for those with ES). Then, at 68% relative humidity, the absorption was much higher, reaching approximately 6% and 34% in mass increase with respect to the initial mass for BF_4_ and ES ILs, respectively, which represented about 27% of the maximum water absorption in the saturated atmosphere for the four samples analyzed here. This could lead us to conclude that the water uptake process is slower for sulfates, which require a much higher humidity grade to incorporate the same proportion of mass as tetrafluoroborates. In contrast, the quantity of water that absorbs the ES with respect to BF_4_ was approximately five times higher in a saturated atmosphere. The most complete study on the evolution of the water mass absorbed with humidity grades is given in ref. [7].

Continuing with the comparison between imidazolium and pyridinium according to Figure 2b, the water desorption process showed a much faster decrease for the BF_4_-based ILs, which started from a lower amount of water. After 50 h of exposure to a dry environment, the decrease in the mass of water contained in the samples was much smoother, reaching Δm values very close to zero for the ILs with the tetrafluoroborate anion. At the end of the experiment, the ILs with the sulfate anion presented Δm values between five and seven times higher, but with such low values that it was not possible to reach definitive conclusions, because Figure 1b shows that the situation was the reverse, although with different cations.

### 3.2. Influence of Surface Area Exposed to Air

The analysis of the results presented in Figure 3a was conclusive because the mass variation experienced by the EMIM-ES samples studied in Petri plates of six different sizes indicated that 100% humidity was a function of the exposed surface area. In addition, Figure 3b shows that the mass of water absorbed per unit area made it clear that the Petri plate with the highest environmental humidity uptake was the one with the smallest area, probably due to edge effects, because of the different adhesion forces established between glass–water and glass–EMIM-ES. To elucidate this point, we took the data for all Petri plates after 136 h in a chamber with a 100% humidity grade and represented them as a function of the plate radius, as shown in Figure 6. We could fit to the points a quadratic curve without a free parameter, i.e.,
(1)$$∆m=A\cdot R^{2}+B\cdot R,$$
where *A* and *B* are the fitting parameters; the first is linked to the water adsorbed on the surface, and *B* is linked to the water absorbed on the perimeter. Thus, we obtain *A* = Δ*m*_S_·π and *B* = Δ*m*_P_·2π, where Δm_S_ and Δ*m*_P_ represent the water absorbed by the surface per square centimeter and the water adsorbed by the perimeter per centimeter, respectively. In the experiment performed using EMIM-ES (shown in Figure 3a), the values of the parameters obtained by fitting the experimental points after 136 h in a moisture-saturated chamber were *A* = 0.2381 g/cm^2^ and *B* = 0.2672 g/cm, which represents Δ*m*_S_ = 75.8 mg/cm^2^ and Δ*m*_P_ = 43.8 mg/cm. Thus, we could estimate the quantity of water adsorbed by the EMIM-ES deposited on a round plate of any radius in a humidity-saturated atmosphere at room temperature.

### 3.3. Influence of Volume

After we knew the influence of the surface area exposed to the atmosphere on water absorption, we performed a new experiment to observe the influence of IL volume on that process. To do that, we deposited in three Petri plates of the same radius 1.0, 1.5, and 2.5 mL of the two ILs previously used, EMIM-ES and EMIM-BF_4_. We left the four samples (plus an empty target Petri plate) in a chamber saturated with moisture (i.e., at 100% relative humidity) and periodically weighed the plates (trying not to shake them). The data obtained are presented in Figure 4, where we observed that a higher IL quantity indicated a higher water absorption for the two ILs studied. We divided the mass of water absorbed by the volume (in mL) of both ILs, and the results are plotted in Figure 7. As observed, while for EMIM-BF_4_, the quantity of water mass absorbed per mL of IL was approximately constant, that did not occur for EMIM-ES, where the quantity of water adsorbed per volume was higher for the sample with lower content and lower for that with a higher volume of IL. To understand these results, we must consider the different characteristics of both ILs with respect to water, because while EMIM-ES was hydrophilic, EMIM-BF_4_ was hydrophobic. Thus, while in the second compound, all the water taken from the atmosphere was mixed into the IL bulk (and thus Δ*m* was proportional to the IL volume), in the first IL, a film of water was formed on its surface, this film thickness being independent of the IL volume. So, we could assume that from the total water taken, the part was mixed with the bulk of the IL (and thus it was proportional to its volume), Δ*m_vol_*, and the other part was deposited onto the IL surface (and so it was independent of the IL volume), Δ*m_film_*. If we write this model in an equation, we obtain the following,
(2)$$∆m=∆m_{vol}\cdot V+∆m_{film}.$$

If we take the data from Figure 4 after approximately 116 h in the water-saturated chamber and apply the above equation, we obtain that the water adsorbed for EMIM-ES at room temperature, in a 3.9 cm diameter plate, in the bulk and the surface was $∆m_{vol}$ = 1.1 g/mL and $∆m_{film}$ = 0.69 g. Performing the same calculation for the EMIM-BF_4_, we obtain that $∆m_{vol}$ = 0.54 g/mL and $∆m_{film}$ = 0.21 g, as expected from the data shown in Figure 4, and while the water adsorbed by the bulk of the IL was roughly double in EMIM-ES than in EMIM-BF_4_, the water forming a film above the IL surface was more than three times higher for the most hydrophilic IL. Thus, it seems that the mechanism of water sorption was different for both ILs, whereas for EMIM-BF_4_, the majority of the water went to the bulk, probably forming water clusters, as described in ref. [15], whereas for EMIM-ES, the water was partially absorbed in the bulk of the liquid, and the other part formed a film on the liquid when the bulk was saturated with water, as described in ref. [14].

### 3.4. Influence of Agitation and Temperature

The data obtained from the experiment performed with the stirred aqueous mixtures of the two ILs indicated the different behaviors of both anions. While ILs with the BF_4_ anion did not lose water, those with ES became nearly dry after 24 h in the chamber. This means that water molecules presented a low mobility in BF_4_-based ILs, and therefore, they could not escape from the bulk of the mixture. In contrast, the intermolecular force of the water molecules in the IL with the ES anion was weak; thus, they exhibited high mobility and could reach the surface of the liquid mixture to escape into the dry atmosphere. These findings are in agreement with previously published results on the molecular interactions of these two ILs with water. Thus, in ref. [36], the diffusion coefficients of the EMIM cation and BF_4_ anion in aqueous mixtures were studied, concluding that water pockets were formed in the bulk, breaking the net interaction formed by the IL ions. Unfortunately, they did not measure the diffusion of the water molecules, but it is expected to be low due to the formation of these water clusters [15,37]. In contrast, in the case of EMIM-ES, water is not incorporated into pockets but uniformly distributed, forming interactions with the EMIM cation and ES anion [38]. Here, they used ATR-IR spectroscopy to deduce that for the mutual interactions among EMIM, ES, and water, the following sequential order of interaction strength was obtained: EMIM−H_2_O−ES > EMIM−ES > ES−H_2_O > EMIM−H_2_O, observing that the interaction of water molecules was weak with both ions. Relating these facts with the water absorption of the moisture experimental data shown in Figure 7, we deduce that EMIM-BF_4_ sorption is mainly superficial, while for EMIM-ES, the bulk of the sample also stores water molecules that travel easily among the IL ions in the bulk.

With respect to the influence of temperature on the moisture absorption process, the quantity of absorbed water increased. This is logical because, at a given relative humidity grade, the atmosphere contains more water at higher temperatures, according to the psychrometric diagram. In any case, the measurements presented here are not complete and must be extended to obtain a clear view of this dependence. The main problem was maintaining a constant temperature in the chamber during the experiment.

## 4. Materials and Methods

### 4.1. Materials

The materials employed in this work were commercial ILs obtained from different suppliers. Table 1 presents the purities, original water content certified by the suppliers, molar mass, and miscibility in water for all of them.

### 4.2. Procedures

All chemicals were handled in an atmospheric chamber to protect them from moisture. The relative humidity was set lower than 15% using silica gel in its interior and by passing a dry air stream through the chamber for 2 h. The relative humidity was measured using a Kestrel 3500 hygrometer (Nielsen-Keller Company, Boothwyn, PA, USA), which ensured an margin of error of 0.1%. Under these conditions, for liquid substances, we placed Petri plates of known diameters in the chamber and poured a given volume of each IL using a pipette. For solid substances, we used a spatula to transfer the chemical from the original container to the Petri plates. Meanwhile, one of the Petri plates was reserved to act as a target and was kept without the IL. All Petri plates were weighed empty and with the ILs (for the target, only empty) with an uncertainty in mass of ±1 mg, and therefore, the mass of each sample for a relative humidity lower than 15% was known. Once these initial preparations were completed, three procedures were performed, as described below:1. Increasing the relative humidity from 15% to 100%: the silica gel was removed from the chamber and water vapor was gradually introduced until 100% humidity was reached. The samples were kept for several days, between 3 and 8 depending on the experiment, avoiding stirring or shaking the samples. The masses of all the samples and the target were recorded on a regular basis throughout the period of exposure to the saturated atmosphere.2. Decreasing the relative humidity from 100% to less than 15%: the samples were then subjected to the reverse process, i.e., no more water vapor was injected, a stream of dry air was passed through the chamber in a controlled manner, and silica gel was introduced to reach the lowest relative humidity values. During this process, the mass of the Petri plates was measured at regular time periods without stirring.3. Increasing the relative humidity in two steps: this was a procedure similar to 1, but at a relative humidity of 68%, the chamber was allowed to stabilize for 24 h, and the Petri plates were then periodically weighed. The steam was then re-injected into the chamber until 100% relative humidity was reached.

## 5. Conclusions

We determined the hygroscopic grade of the ILs EMIM-Cl, EMIM-Br, EMIM-MS, EMIM-ES, EMPY-ES, EMIM-Ts, EMIM-BF_4_, BMIM-BF_4_, dDMIM-BF_4_, BMPY-BF_4_, BMPIPE-TFSI, MPPYRR-TFSI, BMPYRR-TFSI, and MtOAM-TFSI at room temperature and atmospheric pressure. The aim was to elucidate the influence of the chemical nature of the compounds, exposed surface area, sample volume, agitation, and temperature. To this end, we exposed the samples to abrupt increases in relative humidity from 15 to 100% for several days in an atmosphere chamber and then proceeded with the reverse process in a gentle manner.

The results obtained show that moisture capture from the air depends primarily on the anion, and that water sorption is a slow and long process. They also show that the sorption of water from air depends on the nature of the IL, especially the anion, with the chloride anion being of particular importance (chloride ≫ alkyl sulfates~bromide > tosylate ≫ tetrafluoroborate). In contrast, ILs based on the TFSI anion proved to be non-hygroscopic. It was also demonstrated that, at least for EMIM-ES, the mechanism of moisture capture is both absorption and adsorption, and that the smaller the exposed surface area, the higher the ratio of mass increment per unit area.

Studying EMIM-ES and EMIM-BF_4_ samples containing different volumes, we observed that a higher IL quantity meant a higher water sorption for the two ILs, and that the quantity of water per volume was higher for the sample with lower content. In addition, water was quantified. It was also demonstrated that the mechanism of moisture capture was both absorption and adsorption, and that the water absorbed by the ES-based IL was twice as high, whereas the water deposited on the surface as a film was three times higher than that of the BF_4_-based IL.

## Figures and Tables

**Figure 1 ijms-25-04229-f001:**
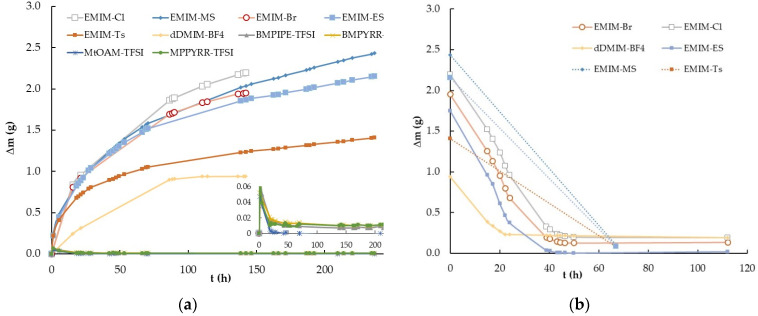
Time-dependent mass variation (Δm vs. t) of the ionic liquids EMIM-Cl, EMIM-Br, EMIM-MS, EMIM-ES, EMIM-Ts, dDMIM-BF_4_, MPPYRR-TFSI, BMPYRR-TFSI, BMPIPE-TFSI, and MtOA-TFSI at room temperature: (**a**) abruptly increasing relative humidity from 15% to 100%; (**b**) decreasing relative humidity from 100% to 15%. Lines are visual guides.

**Figure 2 ijms-25-04229-f002:**
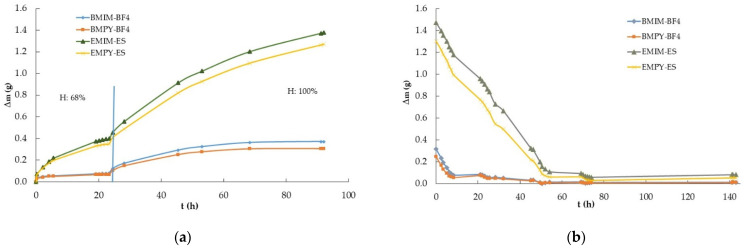
Time-dependent mass variation (Δm vs. t) of ionic liquids BMIM-BF_4_, EMIM-ES, EMPY-ES, and BMPY-BF_4_ at room temperature: (**a**) increasing relative humidity in two steps, from 15% to 68% and from 68% to 100%; (**b**) decreasing relative humidity from 100% to 15%. The vertical blue line separates each step at 68% relative humidity. Lines are visual guides.

**Figure 3 ijms-25-04229-f003:**
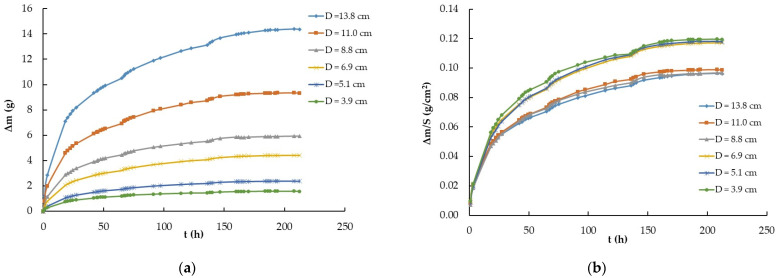
(**a**) Time-dependent mass variation (Δm vs. t) of the ionic liquid EMIM-ES at room temperature as the relative humidity abruptly increased from 15% to 100% for samples contained in Petri plates with diameters of 3.9, 5.1, 6.9, 8.8 11.0, and 13.8 cm; (**b**) time-dependent mass variation per square centimeter (Δm/S vs. t) of the ionic liquid EMIM-ES at room temperature as humidity abruptly increased from 15% to 100% for samples contained in Petri plates with diameters of 3.9, 5.1, 6.9, 8.8, 11.0, and 13.8 cm. Lines are visual guides.

**Figure 4 ijms-25-04229-f004:**
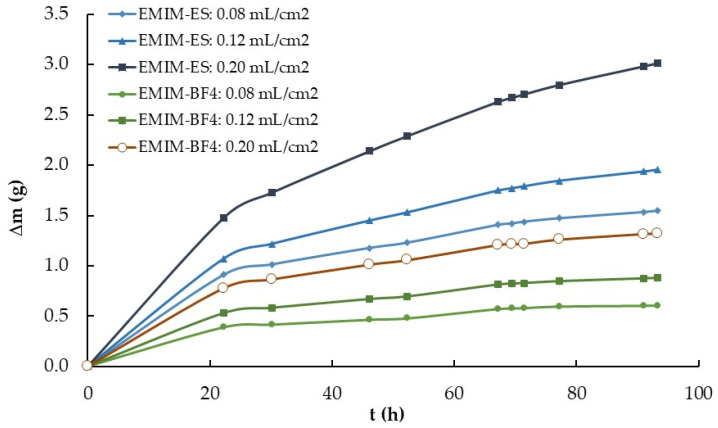
Time-dependent mass variation (Δm vs. t) of two series of 1.0, 1.5, and 2.5 mL of EMIM-ES and EMIM-BF_4_ contained in 3.9 ± 0.2 cm diameter Petri plates. The volume per area for each ionic liquid was approximately 0.08, 0.12, and 0.20 mL/cm^2^. Lines are visual guides.

**Figure 5 ijms-25-04229-f005:**
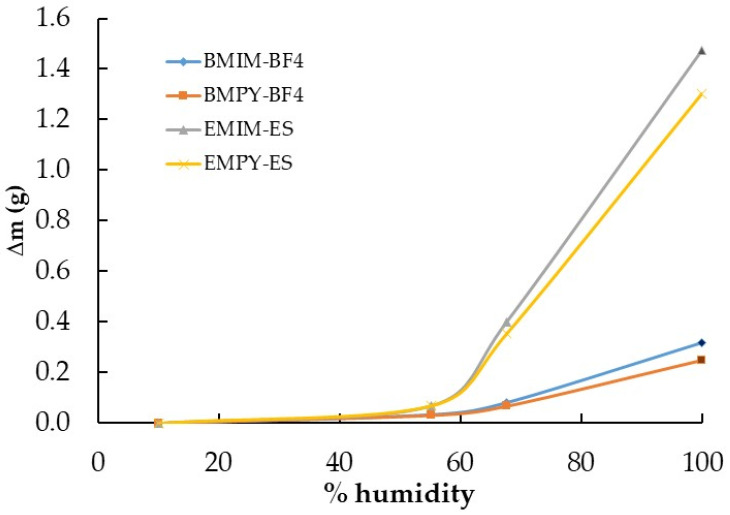
Variation in mass with relative humidity (Δm vs. % humidity) of the ionic liquids BMIM-BF_4_, EMIM-ES, EMPY-ES, and BMPY-BF_4_ at room temperature. Lines are guides to the eye.

**Figure 6 ijms-25-04229-f006:**
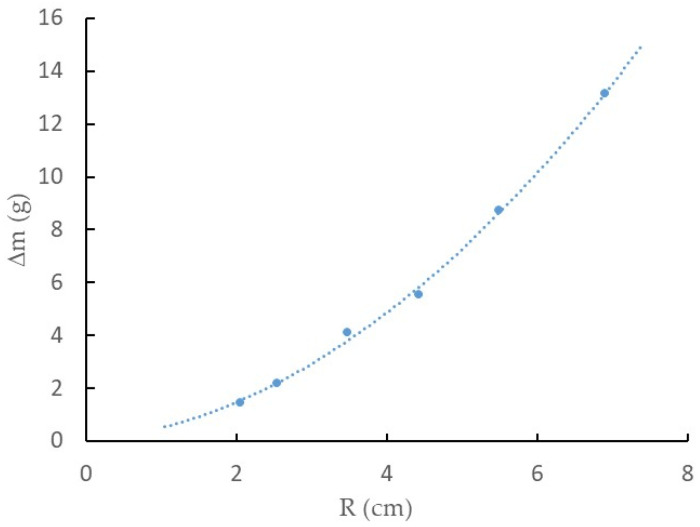
The relationship between the water mass absorption and Petri plate radius for EMIM-ES at room temperature left for 136 h at a relative humidity of 100%. The line is the best fit of a quadratic curve without an independent term (see text for details).

**Figure 7 ijms-25-04229-f007:**
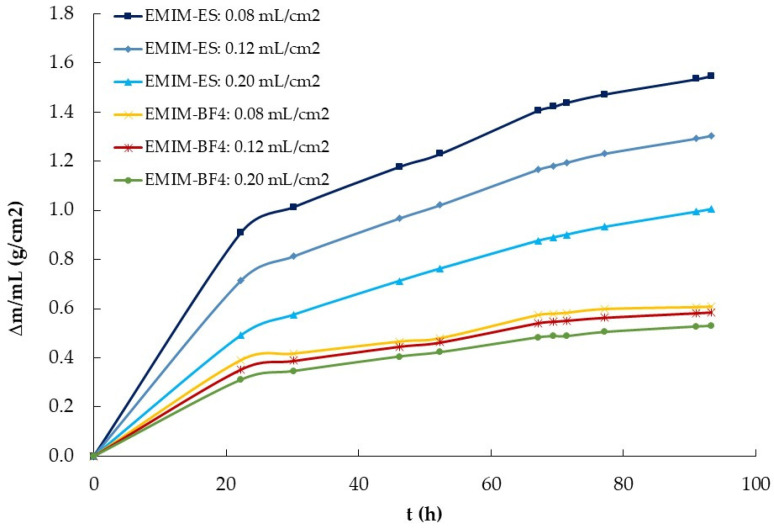
Water mass absorption normalized by the volume of EMIM-ES and EMIM-BF_4_ at room temperature, left at a relative humidity of 100% in Petri plates of the same radius. Lines are guides to the eye.

**Table 1 ijms-25-04229-t001:** Description of the ionic liquids used in this work as follows: source, purity, original water content, molar mass, state of matter at room temperature, and miscibility in water.

IL	Supplier	Purity (%)	Water (ppm)	MW (g/mol)	State	Miscible
EMIM-Cl	Solvent Innovation	99.7	<300	146.62	Solid	Yes
EMIM-Br	Fluka	≥97	-	191.07	Solid	Yes
EMIM-MS	IoLiTec	99	<500	222.26	Solid	Yes
EMIM-ES	IoLiTec	98	<500	236.29	Liquid	Yes
EMPY-ES	Solvent Innovation	99.8	472	247.31	Liquid	Yes
EMIM-Ts	Solvent Innovation	≥99	835.4	282.36	Liquid	Yes
EMIM-BF_4_	IoLiTec	>98%	<1000	197.97	Liquid	Yes
BMIM-BF_4_	IoLiTec	>99	<250	226.02	Liquid	Yes
dDMIM-BF_4_	IoLiTec	98	<1000	338.24	Liquid	No
BMPY-BF_4_	Solvent Innovation	99.9	672.2	237.05	Liquid	Yes
BMPIPE-TFSI	IoLiTec	99	<100	436.44	Liquid	No
MPPYRR-TFSI	IoLiTec	99	<100	408.38	Liquid	No
BMPYRR-TFSI	IoLiTec	99.5	<100	422.41	Liquid	No
MtOAM-TFSI	Solvent Innovation	99	139	648.85	Liquid	No

## Data Availability

Data are contained within the article.

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
