# Peer review of "New Insights into the Hygroscopic Character of Ionic Liquids: Study of Fourteen Representatives of Five Cation and Four Anion Families"

_ijms, 2024, doi:10.3390/ijms25084229_

Round 1

Reviewer 1 Report

Comments and Suggestions for Authors

After carefully reviewing the manuscript “New insights into the hygroscopic character of ionic liquids”, I find it to be both interesting and informative, offering valuable insights. However, before publishing, I recommend minor revisions from the authors to ensure clarity and coherence. Overall, it has great potential to contribute meaningfully to its field. I have provided detailed feedback and suggestions for improvement in the review below.

1.       The experiments are well planned and described. All but one ionic liquids used have purities 98% and above. EMIM Cl however has purity >93%. Can the authors provide an explanation as to what impurities can be present? It is rather unusual to EMIM Cl to have such low purities, since it is solid at room temperature, and the work-up and crystallization is rather simple and effective. I would recommend for further studies to either purchase this compound with higher purity or do recrystallization.

2.       Have the authors considered the influence of densities on the sorption properties of the investigated ionic liquids? For example, EMIM ES has density of 1,24 g/cm3 and EMIM TFSI has density of 1,52 g/cm3.

3.       In addition, EMIM Cl and EMIM Br are solids at room temperature, yet the authors do not mention that anywhere, stating in line 366 and 367 that all the samples were poured onto the petri dishes (“and pour a given volume of each IL in them using a pipette”). This needs explanation.

4.       I believe that the water content should also be confirmed by the Karl-Fisher titration (not just for one sample), not just solely on the mass uptake or decrease. Especially those measurements should be made before using the ionic liquids, not relying on the information provided by the supplier.

Comments on the Quality of English Language

The manuscript should be checked for some minor grammar and spelling mistakes.

Reviewer 2 Report

Comments and Suggestions for Authors

The present work is devoted to the study of hygroscopicity of ionic liquids. The hygroscopicity of ILs is an extremely important parameter that has a significant influence on the manifested physicochemical properties and their further application. Despite this, there are a number of comments on the paper presented below:

1) The title of the paper seems too generalized, considering that it is about specific 14 ionic liquids. At the same time, half of them refer to imidazolium ILs.

2) The Introduction section needs revision. Very little literature was utilized by the authors. As it stands, the introduction is a description of previous studies, i.e. self-citation. In addition, it is very questionable and immodest to indicate the number of citations of their previous works.

3) There is a lot of repetition in the paper. In particular, information about the importance of this study is duplicated in several sections.

4) The paper shows how strongly the anion affects the hygroscopicity parameter. In this case, if an IL with a more hydrophilic cation were used, the cation would have a stronger effect on the hygroscopicity. For example, proton ionic liquids containing hydroxyalkyl groups (triethanolammonium, diethanolammonium, etc.). Only one ammonium IL with a long hydrophobic alkyl chain was used in this work.

5) Section 2.4 and Section 3.5. It is not quite clear on the basis of which experimental data the conclusions presented in the paper were drawn.
